# SELECT: Search-Enhanced Language Models for Analog Circuit Topology Generation

## Abstract

Automating analog circuit topology design is essential to reduce the extensive manual effort required to meet increasingly diverse and customized application demands. Recent advances have applied sequence-to-sequence fine-tuning on pretrained language models to directly generate circuit topologies from user specifications in a single pass. However, these one-shot generation methods failed to generate complex circuits due to their exponentially growing search spaces and limited training datasets. In this paper, we present **SELECT**, a search-enhanced language model framework that integrates simulator-guided Monte Carlo Tree Search (MCTS) with transformer-based decoding to use test-time computation for improved performance. SELECT introduces novel structural token pruning and P-UCB-based node selection to leverage next-token probability distributions to guide the search process. By combining pretrained priors with simulator feedback at inference time, SELECT converges faster than prior search methods and achieves significantly higher generation success rates, improving by up to 435% over RL-based search and 145% over LaMAGIC under a strict tolerance of 0.01. These results establish SELECT as the first scalable framework for complex analog topology generation and a practical step toward LLM-driven circuit design automation. Code and data are available below [1].

## 1 Introduction

Analog circuit topology design sits at the heart of modern electronic systems, enabling everything from efficient power conversion to high-speed signal processing. As device requirements proliferate, varying voltage-conversion ratios, efficiency targets, and performance specifications, the burden on designers to craft bespoke topologies grows heavier. Traditional workflows remain largely manual, demanding extensive domain expertise and hundreds of simulation iterations per new requirement, which in turn prolongs development cycles and delays time-to-market. To meet these challenges, automating the topology design process has become essential: by embedding search and learning methods directly into the design flow, engineers can rapidly explore vast design spaces, reduce iteration counts, and accelerate the creation of optimized analog circuits.

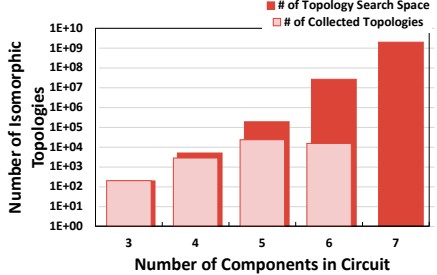

Figure 1: Exponential growth of topology space for circuits with four component types. As the number of components increases, the possible isomorphic topologies expand combinatorially Fan et al. (2024), while the set of practically collected topologies Chang et al. (2024) remains limited, leading to an increasingly sparse coverage of the design space.

Early works (Fan et al. (2021); Zhao and Zhang (2022); Lu et al. (2023)) leverage reinforcement learning (RL) or Bayesian optimization to discover valid topologies using simulation feedback. These methods reduce evaluation costs and produce functional designs, but suffer from two key limitations: (1) they must restart search or retrain policies for every

---

[1] anonymous code release.

new specification, and (2) they have only demonstrated success on small, relaxed 3–5 component circuits. Without access to prior knowledge across tasks, such methods remain inefficient and unscalable for practical usage.

LaMAGIC (Chang et al. (2024)) reframed topology generation as a sequence-to-sequence problem for autoregressive language models, introducing several text-based circuit formulations. Trained on a corpus of 132k 345 component converter topologies, LaMAGIC achieved strong results within this regime. However, as Figure 1 illustrates, the topology search space grows exponentially with component count. At six components, a good dataset coverage becomes impractical: with an average simulation time of 9 seconds per topology, enumerating the full design space would require over 2000 CPU-days. Consequently, LaMAGIC Chang et al. (2024) struggles to transfer knowledge from 345 components to six, and scaling further to complex circuits with 8–10 components becomes infeasible. This gap underscores the need for a new paradigm: one that integrates search as a core component of generation and enables efficient dataset collection for higher-complexity circuits. Building on this insight, our work aims to answer the research question: **how can analog topology generation scale beyond small 3–5 component classroom-level circuits to complex designs with 8–10 components?**

In this work, we introduce **SELECT**, a search-enhanced framework for automated analog topology generation that integrates simulator-guided Monte Carlo Tree Search (MCTS) with pretrained language-model decoding. To the best of our knowledge, SELECT is the first to incorporate search-based decoding into analog circuit generation. We develop a novel MCTS algorithm based on a text-based circuit representation. Our method integrates MCTS to text-based circuit representations by leveraging the LM's beam search and next-token probabilities to guide expansion, rather than exploring the design space uniformly. At each step, we restrict expansion to the top-$k$ most probable tokens, shrink redundant structural tokens, and employ a lookahead planner with simulator feedback whose rewards are backpropagated through the tree. By using the transformer's learned priors and the simulator feedback, SELECT converges faster than the prior search-based methods and leads to higher generation success rates than standard decoding methods. Experimental results show that SELECT achieves a 435%, 145% higher success rate under a low tolerance of 0.01 compared to an RL-search method Fan et al. (2021) and LaMAGIC Chang et al. (2024) with sampling and filtering at the same search budget. Our work also, for the first time, shows the scalability challenge on 7,8,9,10 component circuits by enhancing the existing dataset to higher complex components.

Beyond performance gains, SELECT also addresses the problem of scalability and demonstrates a path to generate complex analog circuit topologies. For the first time, we extend analog topology generation to 7–10 component circuits, enhancing the existing corpus with a large, high-quality collection of six-component topologies obtained through our MCTS-based framework. This unprecedented dataset substantially broadens coverage of realistic analog circuits, establishing the largest and most complex benchmark to date. By enabling generation on circuits well beyond the 3–5 component range of prior datasets Chang et al. (2024), SELECT demonstrates clear scalability trends toward higher-complexity, real-world analog topologies, setting a new standard for practical analog design automation.

## 2 PRELIMINARIES

### 2.1 ANALOG TOPOLOGY DESIGN

In this work, we address the same problem as LaMAGIC Chang et al. (2024): generating customized power converters that meet specific voltage conversion ratios and efficiency targets.

The **voltage conversion ratio** is the output-to-input voltage ratio, while power conversion **efficiency** is the output-to-input power ratio. The **duty cycle** (a number between 0-1) is a design parameter, which controls the ON time of switches, affecting performance. We use five discrete duty cycles: $0.1, 0.3, 0.5, 0.7, 0.9$.

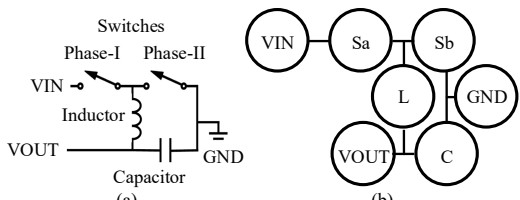

Figure 2: (a) An example power converter circuit and (b) its corresponding graph representation. (Chang et al. (2024)).

We represent circuits as hypergraphs $G$ with vertices $V$ and hyperedges $E$. Vertices include three terminals (input $V_{\text{IN}}$, output $V_{\text{OUT}}$, and ground GND) and four component types (capacitors $C$,

inductors $L$, and switches $S_a$, $S_b$). Hyperedges define connections between components and terminals. Figure 2 shows an example converter with its hypergraph representation.

**Problem Statement**: Given vertices $V$, target conversion ratio $r$, and efficiency $\eta$, our model generates connections $E$ and selects duty cycle $s$ to create a circuit meeting both performance requirements.

## 2.2 SEARCH-BASED METHODS

Previous search-based methods primarily rely on reinforcement learning (RL) and Monte Carlo Tree Search (MCTS) tailored to specific circuit tasks. For example, Fan et al. (2021) models power converter design as a sequential decision process using a UCT-based RL tree with physics-guided pruning, achieving up to 67% fewer SPICE calls than genetic or random search. Zhao and Zhang (2022) applies deep RL to op-amp synthesis, combining symbolic analysis and memorization to converge to feasible designs within hours, significantly faster than graph-grammar engines, but requiring retraining for each circuit class. Lu et al. (2023) integrates variational autoencoders with Bayesian optimization, where BO serves as a principled search strategy to identify spec-compliant topologies more efficiently than traditional methods. Despite these advances, existing approaches still require a fresh search or policy retraining for every new specification, underscoring the need for a more reusable and generalizable generative solution.

## 2.3 LANGUAGE MODEL-BASED METHODS

AnalogCoder (Lai et al. (2024)) leverages prompt engineering in task-agnostic LLMs to iteratively optimize circuit designs through simulation feedback. However, it lacks the ability to tailor formulations for direct specification-to-topology mapping. To address this limitation, LaMAGIC (Chang et al. (2024)) employs SFT with custom circuit formulations to achieve precise specification-to-topology generation. LaMAGIC represents circuit topologies as hypergraphs where components form nodes and their connections form edges. Similarly, CktGNN (Dong et al. (2023)) employs adjacency matrices together with a graph variational autoencoder (VAE) to generate analog topologies like operational amplifiers. AnalogGenie (Gao et al. (2025)) uses Eulerian circuits representation and focuses on optimizing analog circuit performance by combining with a genetic sizing algorithm. However, these approaches remain confined to one-shot generation, relying on limited training data and handcrafted knowledge distillation. As circuit complexity increases, the design space grows combinatorially, making one-shot strategies insufficient. In this work, our SELECT method advances analog topology generation along a disentangled direction: it leverages text-based formulations from prior works but uses the learned model knowledge to guide an inference-time search process, enabling scalable exploration of complex design spaces and achieving higher success rates and performance.

## 3 COMPLEX POWER CONVERTER TOPOLOGY DATASET CONSTRUCTION

The lack of sufficiently large analog circuit datasets continues to hinder the development of AI-based generative methods that aim to automate the design of analog ICs. Some datasets have been presented in previous works, such as AnalogGenie Gao et al. (2025), Align Kunal et al. (2019), CktGNN Dong et al. (2023), and AMSNet Tao et al. (2024). However, they focus on covering diverse circuits with only thousands of examples per circuit type. This limited sample size prevents models from learning internal dynamics of complex circuits. Instead, we focus specifically on power converters, emphasizing complex circuit topologies to help models acquire deeper analog knowledge. However, scaling to higher-component circuits presents several challenges: (1) more components require larger training datasets due to increased task complexity, (2) more components increase the likelihood of generating invalid or useless topologies(e.g. low efficiency circuit), and (3) random arrangements become increasingly unlikely to produce efficient converters as complexity grows.

To address this gap, we make two contributions. First, to demonstrate scalability, we extend the data collection pipeline to build 7–10 component datasets, establishing the largest and most complex corpus of analog power converter topologies to date. This unprecedented dataset substantially broadens coverage of realistic analog circuits, setting a new benchmark for generative analog design research. Second, we construct a dataset of 350k six-component power converter topologies, in addition to the 120k 3–5 component circuits from LaMAGIC Chang et al. (2024), by leveraging the

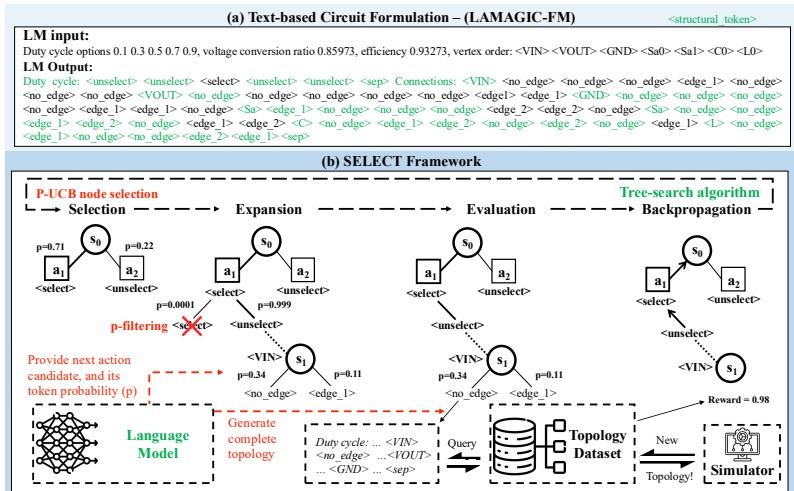

Figure 3: (a) A circuit example of float-input adjacency-based matrix formulation for edge generation task, highlighting its inefficiency due to structural tokens. (b) Illustration of the SELECT framework pipeline through a step-by-step example of leveraging an MCTS algorithm to guide the Transformer generation for analog circuit topology.

trained LaMAGIC model together with our SELECT framework to collect a high-quality dataset. The advantages of this search-based data collection pipeline are detailed in Section 6.4.

## 4 ANALYSIS OF EXISTING TEXT-BASED CIRCUIT FORMULATION

LaMAGIC Chang et al. (2024), a language model-based topology generation framework for automated analog circuit design, introduced three text-based formulations for circuit generation. Among these, the float-input adjacency-matrix formulation (FM) achieved the best MSE results on 6-component circuits. This section examines the use of FM in our search-based methods and analyzes its corresponding challenges.

**Float-input adjacency-matrix formulation (FM).** As shown in Figure 3 (a), FM represents circuit connections as an adjacency matrix for hypergraph, where rows and columns are indexed based on the vertex order given in the input. Distinct tokens <no edge>, <edge 1>, <edge 2>, and <both edges> represent the presence or absence of connections between vertices. While this formulation helps the model to learns better and generalize well in complex circuits, it brings challenges to integrate this method in a search-based framework.

**Limitations.** (1) **Incompatibility with traditional search methods**: Conventional search algorithms traverse circuit topologies by incrementally connecting components with edges. In contrast, FM and other adjacency-matrix formulations operate differently. At each decoding step, they provide probabilities for only four edge connection types between component pairs. This fundamental mismatch means traditional search heuristics cannot be directly applied, necessitating a new search algorithm for search-enhanced language model generation.

(2) **Redundant structural tokens**. Structural tokens are the tokens that are used to maintain the circuit formulation legality without introducing any changes to the target circuit topology. As highlighted in Figure 3 (a), more than half of the tokens in FM formulations are structural tokens. While this formulation generalizes well in complex circuits, it introduces significant inefficiencies in search-based methods.

## 5 SEARCH-ENHANCED LANGUAGE MODEL FRAMEWORK

As illustrated in Figure 3 (b), our SELECT framework integrates MCTS with transformer-based LMs to enhance circuit generation. This approach leverages simulator feedback to guide the search process while employing a Probability-guided Upper Confidence Bound (P-UCB) algorithm that incorporates

LLM token probabilities to balance exploration and exploitation during node selection. To address the inefficiency caused by structural tokens in the circuit formulation, we introduce p-filtering, a novel technique that prevents structural tokens from consuming additional search budget. In the following sections, we detail each component of our framework and explain how they work together to produce an efficient search-guided language generation system for circuit design.

## 5.1 MCTS-based token generation

We propose a transformer generation algorithm that integrates with Monte Carlo Tree Search (MCTS) to perform lookahead planning over partial circuit topologies. While the tree-search structure alone is not efficient enough to tackle the high search space that exists in analog topology, so the traditional beam search algorithm and the token probability suggested by the pre-trained language model served as a good guide for the next exploration data point to guide the search process.

The overall procedure is summarized in Algorithm 1 and visualized in Figure 3. In the following sections, we detail how transformer-learned token probabilities are integrated into each phase of the MCTS process to enable efficient and informed structural exploration in analog topology generation.

**Selection.** We use an Upper Confidence Bound (UCB) strategy to choose which node to explore next,

---

**Algorithm 1** MCTS-based token generation

**Require:** root: initial state; $c$: UCB exploration parameter;
 1: $k$: max children per node; $b$: beam search width;
 2: $p$: threshold for structural token filtering
**Ensure:** Best sequence from MCTS
 3: Initialize tree with root node
 4: **for** $i = 1$ to $max\_rollouts$ **do**
 5:    $node \leftarrow root$              ▷ Selection
 6:    **while** $node$ has children **do**
 7:       $node \leftarrow$ Select child using UCB
 8:    **end while**
 9:    **while** len(node.top-p tokens)==1 **do**    ▷ p-filtering
10:       $node \leftarrow$ CONCAT(node,$next\_tokens$)
11:    **end while**
12:    $next\_tokens \leftarrow$ top-$k$ tokens       ▷ Expansion
13:    **for all** $token \in next\_tokens$ **do**
14:       Add new child node for $token$ to tree
15:    **end for**
16:    $sequence \leftarrow$ Perform beam search    ▷ Evaluation
17:    $r \leftarrow$ Obtain reward via simulation
18:    Backpropagate(node,$r$)       ▷ Backpropagation
19: **end for**
20: **return** sequence with highest reward

---

balancing exploitation and exploration via a tunable parameter $c$. A higher $c$ encourages broader search. Our variant incorporates token probabilities from the language model to guide selection toward likely and underexplored continuations.

**Expansion.** After selecting a node, we first apply **p-filtering** to check whether the top-$p$ distribution is dominated by structural tokens, which often exhibit high top-1 probabilities. If so, we concatenate them directly to avoid consuming rollout budget on uninformative branches. Once non-structural tokens are available, we apply **top-$k$ sampling** to generate multiple child nodes, each representing an extended partial topology.

**Evaluation.** Since partial topologies cannot be directly simulated, we use beam search to complete the sequence from the current node, using a predefined prefix and beam width $b$. The completed topology is then evaluated with NGSPICE to obtain circuit-level performance metrics such as output voltage and efficiency. These are used to compute a reward, which is assigned to the node and backpropagated to update the values of all its ancestors in the tree.

# 6 Experimental results

## 6.1 Experiment setup

**Baseline algorithms.** We compare SELECT with four decoding baselines. **Greedy** denotes the one-shot generation method applying in LaMAGIC. **Beam Search** uses Transformer beam search (beam size 20) without any simulator feedback. **Sampling and filtering (S+F)** generates a set of topologies using the Transformer sampling algorithm (temperature 1.2). Then, it simulates each topology to measure $v_{\text{out}}$ and efficiency, and the candidate closest to the target is returned. To avoid selecting

invalid tokens to break the circuit formulations, we use the top-k sampling with k=3, meaning at each decoding step the transformer only samples from the three most likely tokens. This mirrors AlphaCode(Li et al. (2022)) implementation as well as a baseline in Zhang et al. (2023). **MCTS-Base** is our MCTS variant that uses UCB for node selection but ignores LLM token probabilities, i.e. it treats all equal-count children uniformly. This isolates the benefit of LLM-guided priors. In addition to different decoding algorithms, we compare with **RL-Search**, a prior work (Fan et al. (2021)) that uses an RL search algorithm for power converter generation. They need to run a simulator for each query to give feedback to the RL engine. We set the query budget to 100 per input specification.

**Models and datasets.** Large amount of data for circuits with a higher number of components can be difficult to obtain. To reuse existing knowledge, similar to the LaMAGIC's setting, we extend models trained with 3,4,5-components to be finetuned with 1k and 32k 6-component circuits and leverage our decoding algorithm to evaluate. We follow LaMAGIC (Chang et al. (2024)) to use an encoder-decoder transformer structure with Flan-T5-base pretrained weights. We add a shared linear layer to replace the word embedding layer for numeric inputs. We train the model via conditional generation to learn the mapping between input-output pairs. Model trained with $n$ samples using FM formulation is denoted as FM-$n$.

Evaluating the full 7k LaMAGIC validation set is computationally expensive and inefficient, as it includes numerous low-performing circuits (with low conversion ratios and efficiencies). Instead, we construct two subsets from the original validation set: **6-comp-easy**, consists of 100 randomly selected samples, and **6-comp-hard**, consists of 100 high-performance samples chosen with highest efficiency across various conversion ratios. A detailed selection decision and visualization of the validation dataset is provided in Section D.2.

**Evaluation metrics.** Our primary evaluation metric is success rate, which is the percentage of generated circuits that satisfy the preset performance target $(v^*, e^*)$. For each target pair $(v^*, e^*)$ and a search budget n, the method generates up to n candidate circuits and retains the best-performing sample. A circuit is considered successful if its simulated voltage conversion ratio and efficiency $(v, e)$, obtained using NGSPICE Nenzi and Vogt (2011), both fall within a tolerance t of the target outputs $(v^*, e^*)$:

$$|v - v^*| \leq t \quad \text{and} \quad |e - e^*| \leq t.$$

We report two variants: (1) **tolerance-based success rate**, with $t \in \{0.01, 0.02, \ldots, 0.10\}$, and (2) **iteration-wise success rate**, which tracks success under strict tolerance $(t = 0.01)$ as the number of generated candidates increases. Candidate circuits that fail to compile or simulate are counted as failures. In addition, we report mean squared errors (MSEs) for voltage conversion ratio and efficiency to quantify deviation magnitudes among valid circuits.

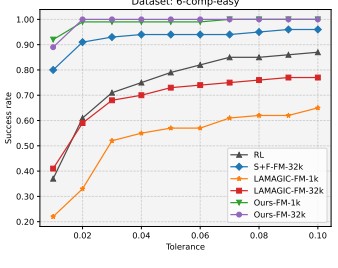

(a) Tolerance sweep on 6-comp-easy.

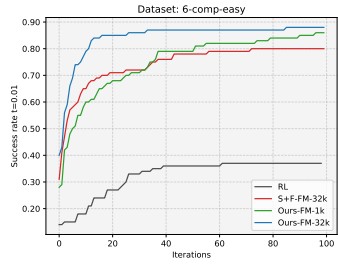

(b) Iteration progression at $t = 0.01$.

Figure 4: **Comparison of success rates on voltage conversion tasks.** We evaluate (i) an RL-search baseline Fan et al. (2021), (ii) one-shot LAMAGIC Chang et al. (2024), and (iii) our MCTS-guided search approach. Experiments are run on the 6-component-easy benchmark, averaging over 100 independent trials. (a) shows how tolerant thresholds affect final success, while (b) tracks convergence speed at a stringent tolerance of $t = 0.01$.

| Method | # Train Data | Success Rate ($t = 0.01$) | | MSE (Voltage) | |
|---|---|---|---|---|---|
| | | 6-comp-easy | 6-comp-hard | 6-comp-easy | 6-comp-hard |
| RL-Search | – | 0.37 | 0.17 | 0.04384 | 0.01331 |
| LAMAGIC | 1 000 | 0.22 | 0.06 | 0.33058 | 0.21891 |
| Sampling + Filter | 1 000 | 0.70 | 0.33 | 0.02162 | 0.00702 |
| MCTS-Base | 1 000 | 0.62 | 0.17 | 0.00441 | 0.01538 |
| Ours (c=4) | 1 000 | 0.89 | 0.52 | 0.00097 | 0.00150 |
| LAMAGIC | 32 000 | 0.41 | 0.18 | 0.19547 | 0.26231 |
| Sampling + Filter | 32 000 | 0.96 | 0.51 | 0.02069 | 0.00388 |
| MCTS-Base | 32 000 | 0.67 | 0.31 | 0.02902 | 0.00824 |
| Ours (c=4) | 32 000 | 0.89 | 0.74 | 0.00005 | 0.00037 |

Table 1: Success rates and MSEs measured on voltage for RL-Search Fan et al. (2021), one-shot LaMAGIC Chang et al. (2024), sampling+filter, MCTS-Base, and our method, at two training-data budgets.

| Method | # Train Data | Success Rate ($t = 0.01$) | | MSE (Voltage) | | MSE (Efficiency) | |
|---|---|---|---|---|---|---|---|
| | | 6-comp-easy | 6-comp-hard | 6-comp-easy | 6-comp-hard | 6-comp-easy | 6-comp-hard |
| Greedy | 1 000 | 0.21 | 0.05 | 0.33 | 0.22 | 0.16 | 0.28 |
| Beam Search | 1 000 | 0.41 | 0.13 | 0.033 | 0.043 | 0.022 | 0.029 |
| Sampling + Filter | 1 000 | 0.70 | 0.20 | 0.0216 | 0.0070 | 0.0053 | 0.0081 |
| MCTS-Base | 1 000 | 0.62 | 0.17 | 0.0044 | 0.0154 | 0.0016 | 0.0093 |
| Ours (c=4) | 1 000 | **0.83** | **0.52** | **0.00016** | **0.00150** | **0.00057** | **0.00135** |
| Greedy | 32 000 | 0.37 | 0.12 | 0.195 | 0.262 | 0.165 | 0.174 |
| Beam Search | 32 000 | 0.51 | 0.32 | 0.025 | 0.027 | 0.013 | 0.023 |
| Sampling + Filter | 32 000 | 0.70 | 0.33 | 0.022 | 0.007 | 0.005 | 0.008 |
| MCTS-Base | 32 000 | 0.67 | 0.31 | 0.029 | 0.008 | 0.005 | 0.010 |
| Ours (c=4) | 32 000 | **0.84** | **0.65** | **0.00006** | **0.00029** | **0.00003** | **0.00019** |

Table 2: Performance at threshold $t = 0.01$ for both 1 k and 32 k training–data budgets. All methods use up to 100 Transformer generations.

## 6.2 GENERATION RESULTS ON 6-COMPONENT CIRCUIT

**Comparison with RL-Search method.** We run the RL-search method (Fan et al. (2021)) for two days to obtain all specifications from our testing set for 6-comp-easy, 6-comp-hard. Since this work only constrains the voltage conversion ratio in topology generation, we evaluate the performance on success rates and the MSE only with voltage conversion ratios. As shown in Figure 4a and Table 1, our search-enhanced methods largely outperform RL-search baselines and the one-shot generation methods, with a success rate of 0.37 (RL) and 0.91 (Ours) when using the FM-32k model on a tight tolerance of 0.01 on dataset: 6-comp-easy.

**Comparison with other decoding algorithms.** For a fair comparison, we evaluate the best topology found by the different decoding algorithms when they use the same number of Transformer generations. The experiments are run on both 6-comp-easy and 6-comp-hard, and the success rate is evaluated on both voltage and efficiency.

Results are shown in Table 2. Our method consistently outperforms all the other baselines on both validation datasets for various tolerance thresholds. The advantages of our methods are more evident on 6-comp-hard dataset where a greater performance gain is observed. Overall, these results confirm that our algorithm indeed generates better topologies for the target voltage and efficiency. Specifically, S+F uses the same number of transformer generations. while their performance is overall outperformed by SELECT. Comparing with MCTS-Base, this confirms that the LLM probability guidance is crucial in the node selection stage. A runtime breakdown for our framework is also provided in Table 5c.

To illustrate the exploration efficiency of various decoding methods, Figure 5b plots the strict-tolerance success rate (t=0.01) as a function of iteration count. For SELECT, we fix the top-k sampling budget at k=3 with a single beam (b=1) and sweep the P-UCB exploration constant c. We observe that c =1 yields faster gains in the very early iterations, while a larger c (c=4) drives

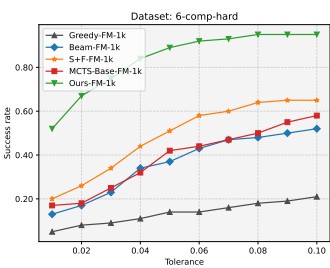

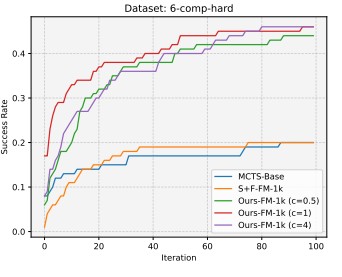

| Metric | Avg Time |
|---|---|
| GPU gen. | 0.19s |
| Sim (no cache) | 10.18s |
| Sim (cached) | 4.10s |
| **Total time** ($n = 100$) | **9.22h** |

(a) 6-component tolerance sweep.

(b) Iteration progression at $t = 0.01$.

(c) Runtime illustration.

Figure 5: **Comparison of success rates and runtime analysis.** (a) Varying error tolerance shows how lenient thresholds impact success rate. (b) Sweeping exploration constant $c \in \{0.5, 1, 4\}$ with pruning probability $p = 0.99$ shows its effect on convergence. (c) Average runtime breakdown and total evaluation time on the 6-comp-hard benchmark for our SELECT framework. For the measured total runtime under a search budget of $n = 100$, early stopping is applied: once a successful target is found at iteration $t$, the search terminates without proceeding to $t + 1$.

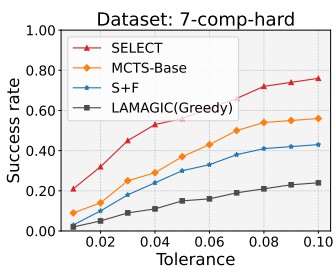

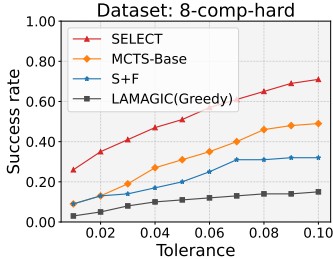

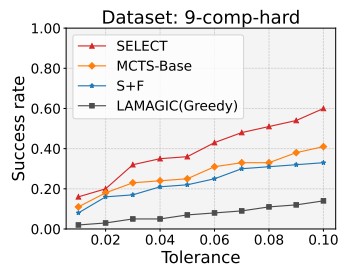

(a) 7-component tolerance sweep.

(b) 8-component tolerance sweep.

(c) 9-component tolerance sweep.

Figure 6: Comparison of generation results on validation sets with varying number of components (7 $\rightarrow$ 9), for Greedy, Sampling + Filtering (S+F), MCTS-Base, and our SELECT methods.

more exploration and ultimately achieves the highest success. In contrast, without the LLM token probability guidance, MCTS Baselines explores less effectively and converges to sub-optimal results with other baseline sampling and filtering.

### 6.3 GENERATION RESULTS ON 7-9 COMPONENT CIRCUIT

To evaluate scalability on complex higher-component circuits, we retrain the baseline with 4K samples (1K each from 6–9 components) and construct 7/8/9-comp-hard validation datasets. Detailed training and validation setup is provided in Appendix C.1 and D.2.

From Figure 6, we observe that as the number of components increases from 7 to 9, all methods experience some performance degradation to generate higher complexity circuits. Nevertheless, our approach consistently outperforms all baselines across the 7/8/9 benchmarks, maintaining high success rates. In particular, structured search methods (SELECT and MCTS-Base) clearly surpass the S+F approaches, underscoring the importance of structured search in graph-generation tasks such as circuit topology design. Table 3 provides the quantitative comparison: our method achieves 10×/13×/4× improvements over the greedy baseline on the 7/8/9 validation sets, and up to 100× lower MSE in voltage targets and 43× lower MSE in efficiency targets.

Overall, the generation results on 7–9 component circuits support our claim that leveraging test-time search enables our method to scale more effectively than baseline strategies as circuit complexity increases, making it particularly suitable for higher-component designs and future large-scale circuits. Results on 10-component circuits are further reported in Section C.2.1, demonstrating the feasibility of generating valid 10-component topologies without any fine-tuning on 10-component data, highlighting the robustness and generalization capability of our approach.

| Method | Success Rate ($t = 0.01$) | | | MSE (Voltage) | | | MSE (Efficiency) | | |
|---|---|---|---|---|---|---|---|---|---|
| | 7-comp-easy | 8-comp-hard | 9-comp-hard | 7-comp-easy | 8-comp-hard | 9-comp-hard | 7-comp-easy | 8-comp-hard | 9-comp-hard |
| Greedy | 0.02 | 0.02 | 0.04 | 0.763 | 0.904 | 0.906 | 0.338 | 0.348 | 0.377 |
| Sampling + Filter | 0.03 | 0.09 | 0.09 | 0.269 | 0.372 | 0.188 | 0.231 | 0.190 | 0.094 |
| MCTS-Base | 0.09 | 0.09 | 0.11 | 0.115 | 0.154 | 0.481 | 0.045 | 0.044 | 0.124 |
| Ours (c=4) | **0.21** | **0.26** | **0.16** | **0.125** | **0.009** | **0.046** | **0.024** | **0.008** | **0.027** |

Table 3: Performance at threshold $t = 0.01$ with varying number of components($7 \rightarrow 9$). All methods use up to 100 Transformer generations.

## 6.4 MCTS AS AN EFFECTIVE DATA COLLECTION METHOD

MCTS combined with generative models represents a powerful yet often overlooked approach for high-quality data collection in complex design spaces. Traditional random generation methods become exponentially ineffective as design complexity grows, with the vast topology search space severely diminishing the probability of discovering valid, high-performance configurations. Our empirical analysis quantifies this limitation: in a random generation of 10,000 6-component circuits, 66.13% exhibited efficiency below 2%, rendering them practically unusable for both application and training purposes. In contrast, our MCTS-based approach significantly mitigates this inefficiency problem,

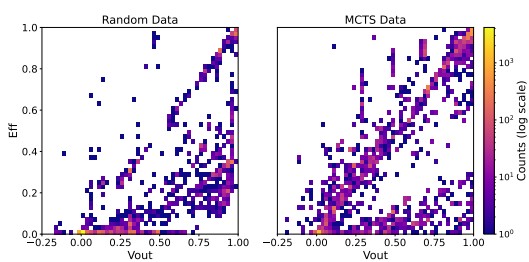

Figure 7: The Vout vs efficiency distribution of our model collected dataset vs. random connection generated dataset.

reducing the proportion of low-performing circuits to just 18.2%. More importantly, our method substantially enhances the discovery of high-quality designs, generating 23.27% of circuits with efficiency exceeding 90%-nearly three times higher than the 8.3% achieved through random generation. The complete efficiency distribution illustrated in Figure 7 demonstrates this substantial quality difference. This shows that beyond immediate circuit applications, our approach can facilitate an efficient mechanism for collecting a high-quality dataset for automatic discovery of unconventional topology and enable further training for the language models.

## 7 CONCLUSION

In this work, we propose SELECT, a search-enhanced language model framework for analog circuit topology generation. SELECT addresses key limitations of prior methods by tightly integrating simulator-guided Monte Carlo Tree Search (MCTS) with transformer-based decoding. Unlike traditional search or language model approaches, SELECT capitalizes on both the learned priors of pretrained models and the dynamic feedback of circuit simulators to navigate vast and complex design spaces. Through a novel adaptation of the MCTS algorithm, we introduce techniques such as P-UCB node selection, top-p expansion, and structural token filtering to better align with the adjacency-matrix-based circuit formulations.

Extensive experiments across varying data regimes and difficulty settings confirm that SELECT achieves significantly higher success rates and lower error metrics compared to prior RL-based search methods and language model decoding strategies like sampling and filtering. Beyond topology generation, we also demonstrate the utility of SELECT as a high-quality data collection engine for scaling analog datasets in low-coverage regimes.

Future work includes extending SELECT to discover more efficient circuits, generalizing to more complex analog designs, and developing search-friendly formulations to reduce structural token overhead for more efficient search.

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

APPENDIX INDEX

# A   ICLR STATEMENTS

## A.1   LLM USAGE STATEMENT

We used Large Language Models (LLMs) solely to polish the writing of this paper, such as improving grammar, clarity, and readability. All technical content, experiments, analyses, and conclusions were conceived, implemented, and validated entirely by the authors without reliance on LLMs.

## A.2   ETHICS STATEMENT

This work focuses on automating analog circuit topology generation through language models and Monte Carlo Tree Search. The research does not involve human subjects, sensitive personal data, or security-critical systems. Our datasets consist entirely of synthetically generated circuits and simulator outputs (NGSPICE), ensuring that no proprietary or private information is used. We adhere to the ICLR Code of Ethics by maintaining transparency in data curation, simulation protocols, and evaluation methodology. We believe this work poses minimal risk of harm and contributes positively to the broader EDA and machine learning communities by advancing open, reproducible research for sustainable semiconductor design automation.

## A.3   REPRODUCIBILITY STATEMENT

We have taken several steps to ensure reproducibility. All datasets, simulation settings, and evaluation subsets (e.g., 6-comp-easy and 6-comp-hard) are fully described in Section 6.1 and Appendix D.2. Implementation details for SELECT, including MCTS integration, node selection strategy, and structural token pruning, are provided in Section 5. Experimental configurations, hyperparameters, and ablation protocols are reported in Section 5. We also provide anonymous supplementary material with source code and scripts for model training and SELECT and evaluation, enabling full replication of our results.

# B   FURTHER EXPLANATION OF METHODOLOGIES

## B.1   P-UCB NODE SELECTION

To guide exploration during tree traversal, we extend the standard Upper Confidence Bound (UCB) Kocsis and Szepesvári (2006) strategy by incorporating the token probability predicted by the language model. The P-UCB score for each child node is computed as:

$$exploration = \sqrt{\frac{log(node.visit\_count)}{child.visit\_count}}, exploitation = child.value$$

$$score = exploitation + c \times exploration * node.token\_probability$$

where $node.visit\_count$ is the record of the number of times a node has been visited. node. token_probability is the token probability of transforming from the decoding of the last token in the existing node. c is the hyperparameter for the exploration term to balance exploration and exploitation in the existing tree search structure. In the **selection** phase, we always select the child with best value. Intuitively, UCB Node Selection function would visit a child more often, if 1) the child has a better node.value, 2) the child has a higher token probability suggested by the transformer, or 3) $parent.visit\_count$ is large while $child.visit\_count$ is low, meaning the child is under-explored. In our experiment settings, we set c as 0.5, 1, 4. In the following steps of Algorithm 1, the algorithm keep calling $UCB\_SELECT$ until we reach a root node. it then **expands** the selected node and **evaluates** the node via simulator feedback. Finally, the reward $r$ is backpropagated to its parent recursively until it reaches the root. The

value mechanism is node.value ← max((node.value),r) for the current node and all of its ancestors.

---

**Algorithm 2** P-UCB node selection

---

**Require:** node: the current node in the MCTS tree
 1:   $c$: exploration parameter
**Ensure:** Selected child node with the highest UCB score
 2: **function** UCB_SELECT($node, c$)
 3:  $best\_score \leftarrow -\infty$
 4:  $best\_child \leftarrow$ None
 5:  **for** $child$ in $node.children$ **do**
 6:    $exploitation \leftarrow child.value$
 7:    $exploration \leftarrow \sqrt{\frac{\log(node.visit\_count)}{child.visit\_count}}$
 8:    $score \leftarrow exploitation + c \cdot exploration \cdot node.token\_probability$
 9:    **if** $score > best\_score$ **then**              ▷ Selection
10:      $best\_score \leftarrow score$
11:      $best\_child \leftarrow child$
12:    **end if**
13:  **end for**
14:  **return** $best\_child$
15: **end function**

---

## C    FURTHER EXPLANATION OF METHODOLOGIES

### C.1    MODEL TRAINING DETAILS AND COMPUTE RESOURCES

The `Flan-T5-base` model consists of 12 transformer layers in both the encoder and decoder. Each layer includes key and value projections with a dimensionality of 64, a feed-forward network with a hidden size of 2048, and employs 12 attention heads. Overall, the model contains approximately 248 million parameters.

To adapt the tokenizer for our specific application, we add custom tokens to its vocabulary. For the SFM task, the following tokens are introduced: `<sep>`, `<duty 0.1>`, `<duty 0.3>`, `<duty 0.5>`, `<duty 0.7>`, `<duty 0.9>`, `VIN`, `VOUT`, `GND`, `Sa`, `Sb`, `C`, `L`, `<no edge>`, `<edge 1>`, `<edge 2>`, and `<both edges>`.

Training is conducted on a machine equipped with eight NVIDIA A5000 GPUs. The language model is trained over 30 epochs using the AdamW optimizer with an initial learning rate of $3 \times 10^{-4}$. A cosine learning rate schedule is applied with 300 warmup steps. The batch size is set to 128, L2 regularization is applied with a strength of $10^{-5}$, and the dropout rate is set to 0.1.

### C.2    ABLATION STUDIES

#### C.2.1    GENERATION RESULTS ON UNSEEN(UNTRAINED) 10-COMP CIRCUIT

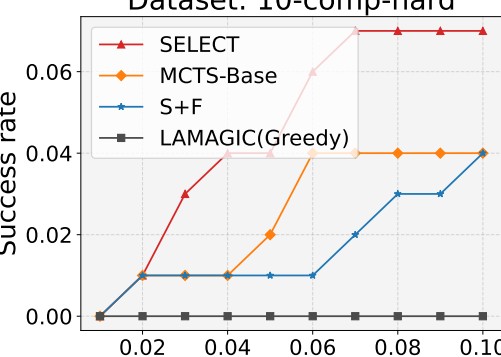

(a) 10-component tolerance sweep.

Figure 8: Generation results on 10-comp circuit validation set, for Greedy, Sampling + Filtering (S+F), MCTS-Base, and our SELECT methods.

To further assess the robustness of our search-based approach, we evaluate it on an unseen, untrained 10-component dataset. As expected, performance degrades and only a few valid circuits are generated. Nevertheless, the results highlight two important observations: (1) even without pre-training, our method is still able to generate higher-complexity circuits, suggesting a clear trend toward scalability across different component counts and circuit types, even when such configurations are absent from the training set; and (2) the relative performance advantage persists in this untrained regime, with SELECT consistently remaining the top-performing method.

#### C.2.2    UCB SELECTION WITHOUT LLM PROBABILITY GUIDANCE

Within our tree structure framework, we incorporate probability guidance during node selection to leverage the LLM's capabilities in directing tree-based sampling. To validate the effectiveness of this LLM-provided probability guidance, we conducted a comparative analysis by modifying the MCTS baseline from standard UCB node selection to our version of P-UCB node selection. Figure 9 confirms that the probability guidance from P-UCB is useful. However, our method still outperforms MCTS(P-UCB) through the implementation of node shrinking, which effectively addresses the challenges

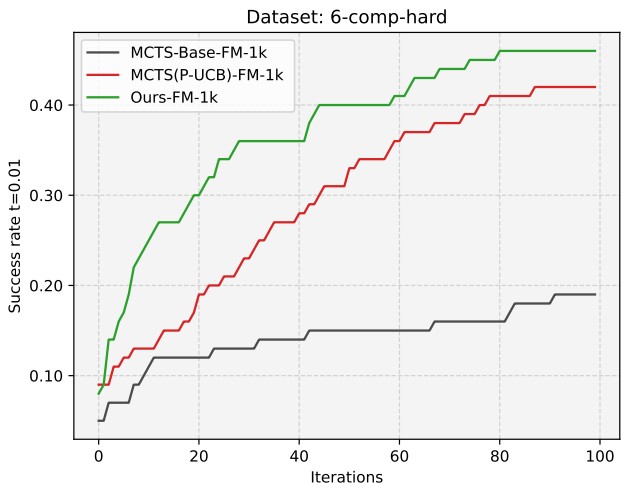

Figure 9: Success rate of our method with MCTS (Baseline) and its MCTS(P-UCB) variants.

posed by structural tokens in circuit formulation. This advantage is particularly pronounced during the early exploration iterations.

# D EXISTING DATASET DISTRIBUTION

## D.1 OVERALL DATASET

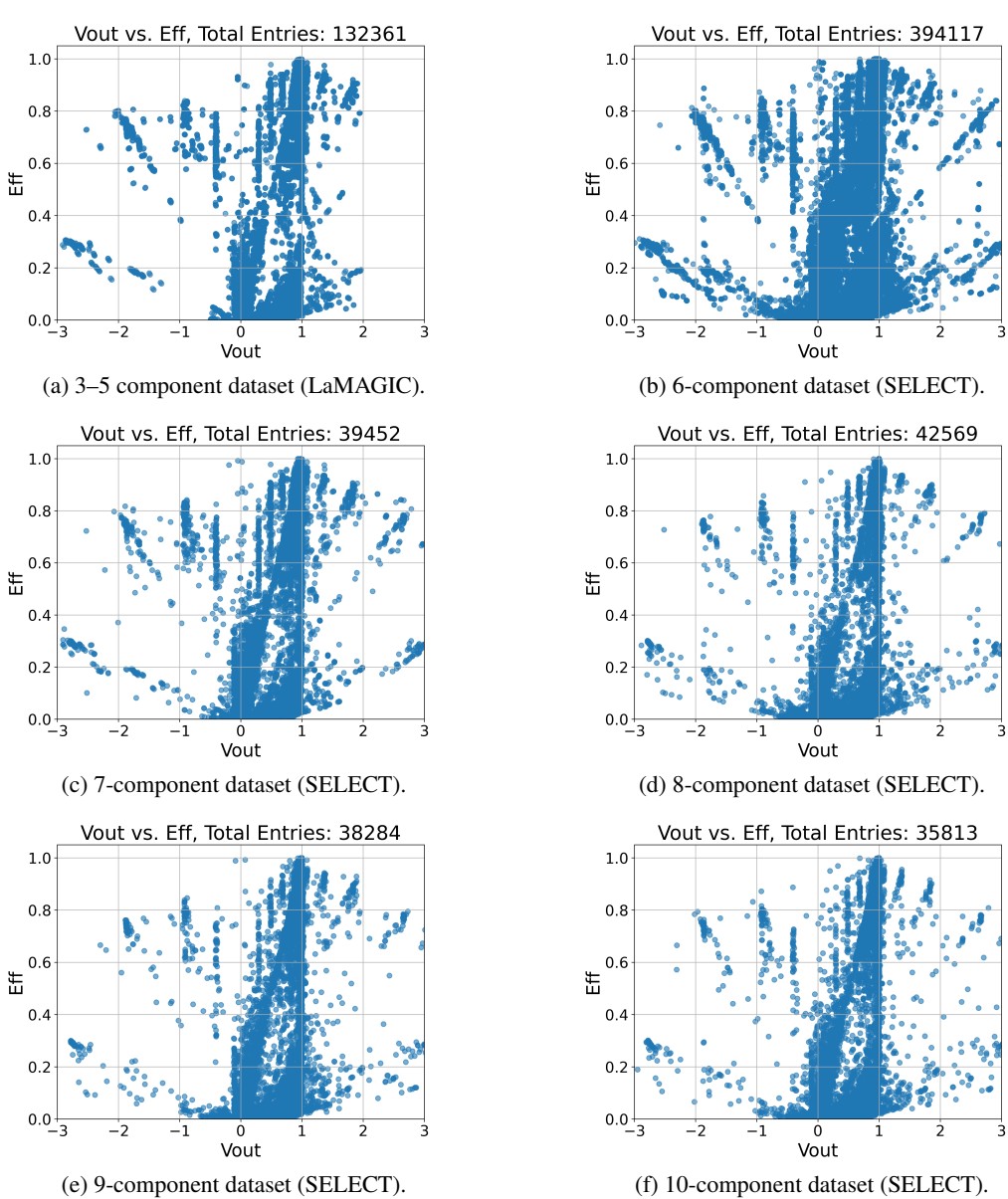

(a) 3–5 component dataset (LaMAGIC).

(b) 6-component dataset (SELECT).

(c) 7-component dataset (SELECT).

(d) 8-component dataset (SELECT).

(e) 9-component dataset (SELECT).

(f) 10-component dataset (SELECT).

Figure 10: Conversion ratio ($V_{out}$) versus efficiency distributions for datasets with 3–9 components.

## D.2 VALIDATION DATASET

**6 component validation:** The **6-comp** dataset (Figure 11a) is the original LAMAGIC 7k validation set. Evaluating the entire set is infeasible: with a search budget of 100, it would require ∼63 days on our A5000 GPUs. And it is also inefficient as it contains numerous low-performing circuits( circuits with extremely low $v$ and low

To address this bottleneck, we construct two 100-sample subsets: 1.) **6-comp-easy**, consisting of 100 randomly selected samples (uniform baseline). 2.) **6-comp-hard**, consisting of 100 high-efficiency samples. Specifically, for $v \in [0.1, 1.1]$ at 0.1 intervals, we select the top-10 most efficient designs, ensuring broad coverage of the majority region.

**7-10 component validation:** For higher-component datasets (**7/8/9/10-comp-hard**), we extend the evaluation to unconventional regions following recommendations from domain experts in analog circuit design. samples with $v \in [-3, 0] \cup [1, 3]$ and high efficiency is more valueable. Specifically, we allocate 20 samples to $v \in [-3, 0] \cup [1, 3]$ (selecting the top-2 most efficient designs per 0.5 interval), and 80 samples to the typical region $v \in [0, 1]$ (selecting the top-8 per 0.1 interval).

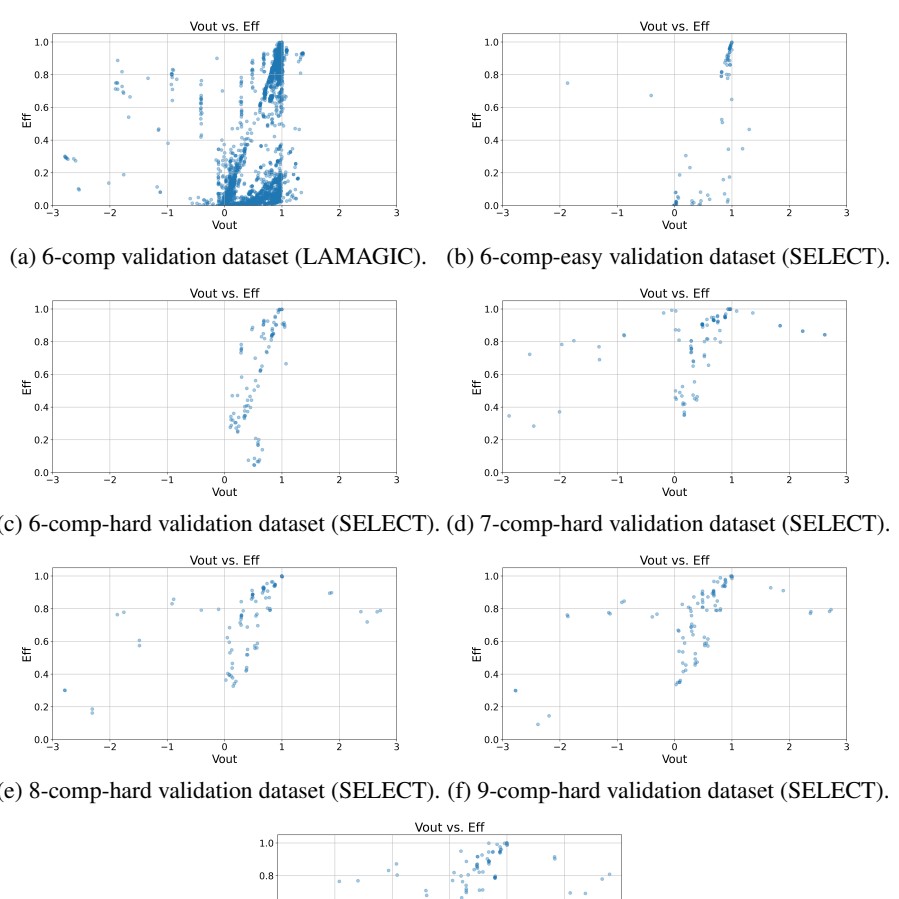

(a) 6-comp validation dataset (LAMAGIC).    (b) 6-comp-easy validation dataset (SELECT).

(c) 6-comp-hard validation dataset (SELECT). (d) 7-comp-hard validation dataset (SELECT).

(e) 8-comp-hard validation dataset (SELECT). (f) 9-comp-hard validation dataset (SELECT).

(g) 10-comp-hard validation dataset (SELECT).

Figure 11: Conversion ratio ($V_{\text{out}}$) versus efficiency distributions across validation datasets with 6–10 components.