# OpenReview forum: "SELECT: Search-Enhanced Language Models for Analog Circuit Topology Generation"
_ICLR.cc/2026/Conference — Submitted to ICLR 2026_

### Official Review · Reviewer_v82m · 2025-10-16

**Soundness:** 2
**Presentation:** 2
**Contribution:** 2
**Rating:** 4
**Confidence:** 5

**Summary:**

This work introduces a search-enhanced framework for automated analog topology generation that integrates simulator-guided Monte Carlo Tree Search (MCTS) with pretrained language-model decoding.

**Strengths:**

SELECT is the first to incorporate search-based decoding into analog circuit generation
SELECT achieves a 435%, 145% higher success rate under a low tolerance of 0.01 compared to an RL-search method Fan et al. (2021) and LaMAGIC Chang et al. (2024) with sampling and filtering at the same search budget.

**Weaknesses:**

1. Clarification needed on circuit complexity claims
The characterization of 8–10 component circuits as "complex designs" may benefit from additional context. As reference:

A basic single-stage cascode op-amp (Design of Analog CMOS Integrated Circuits, 2nd Ed., p. 350, Fig. 9.8) uses 8 components.
The classical two-stage op-amp from Ahuja et al. (1983) uses approximately 20 components (Fig. 3b) (An improved frequency compensation technique for CMOS operational amplifiers).
More recent designs, such as the switched op-amp from Young-Ju et al., use 34 components excluding biasing and CMFB (Fig. 5) (A 12 bit 50 MS/s CMOS Nyquist A/D Converter With a Fully Differential Class-AB Switched Op-Amp).

It would be helpful if the authors could clarify their definition of "complex" and provide comparative context relative to typical analog circuit design practice.

2. Design space analysis requires more detailed justification
Figure 1 presents a design space showing 100 topologies for 3-component circuits and 10,000 for 5-component circuits. To better understand this contribution, the authors should address:

What proportion of these topologies are functionally viable?
How many represent meaningful design variations worth including in the training set?
What criteria were used to filter the design space?

3. Novelty claim needs supporting evidence
The claim that this work is "the first time" analog topology generation extends to 7–10 component circuits would benefit from a more comprehensive literature review. Specifically, it would be helpful to see:

A systematic comparison with prior work (AnalogCoder, Lamagic, CktGNN, AnalogGenie, etc.)
Documentation of the maximum circuit complexity handled by these earlier methods
Clear delineation of what constitutes a substantive advance beyond prior work

4. Relationship to existing MCTS-LLM methods needs clarification
MCTS has been widely adopted in LLM-based code generation (e.g., "Planning with Large Language Models for Code Generation," "Large Language Models as Commonsense Knowledge for Large-Scale Task Planning"). The paper would be strengthened by:

Discussing how the proposed approach differs from or builds upon these methods
Clarifying domain-specific adaptations required for analog circuit generation
More explicitly articulating the unique contributions beyond applying established MCTS-LLM techniques

5. Reproducibility concerns
The absence of supplementary materials (code, datasets, or detailed implementation specifications) limits reproducibility. Providing these resources would significantly strengthen the contribution and enable the community to build upon this work.

**Questions:**

see weakness

---

> ### Author Response · Authors · 2025-12-03
>
> [W1 & W3] We appreciate the reviewer’s comment and acknowledge that our original wording may have overstated the notion of “complex.” In our context, we use “complex” to characterize circuits with a larger number of components relative to what existing one-shot generation methods (e.g., LaMAGIC) can reliably produce. Under the LaMAGIC formulation, one-shot generation is largely constrained to simple, low-component topologies. In contrast, our framework substantially extends this capacity and enables the successful generation of circuits with significantly more components.
> Regarding comparisons to AnalogGenie, our approach is fully compatible with their formulation, and there is no conceptual barrier to integrating our method. However, AnalogGenie does not open-source its simulation backend nor key evaluation metrics (e.g., FoM definitions), which prevents faithful reproduction of their results and thus precludes a meaningful comparison.
> As AnalogCoder and CktGNN do not focus on topology generation, we will not compare with them directly.
>
> [W2] The criteria for including a circuit in the training set are: (i) the circuit must be simulatable and functional, and (ii) its topology must be isomorphic to an existing graph in the database.
>
> [W4] Our method incorporates several domain-specific adaptations tailored to analog circuit topology generation. For example, we introduce p-filtering to remove structural tokens that do not contribute to topological reasoning, thereby improving both training stability and generation quality.
>
> [W5] The code implementation has been open-sourced via the anonymous link.

---

### Official Review · Reviewer_ahJq · 2025-10-30

**Soundness:** 3
**Presentation:** 3
**Contribution:** 2
**Rating:** 4
**Confidence:** 4

**Summary:**

This work aims at improving the performance of automatic analog circuit topology design.
The main challenge of existing approaches is that they fail to generate complex circuits due to the exponentially growing search spaces and limited training datasets.
The main idea of this work is to leverage test-time computation to improve performance.
Specifically, it introduces token pruning, leverage next-token probability distribution to guide the search process, and uses simulator feedback at inference time.

**Strengths:**

- The formulated problem is practical and valuable, which aims to generate complex analog circuit topology with more than six components.
- The proposed method is reasonable and makes sense.
- The writing is clear and easy to follow.

**Weaknesses:**

- In lines 157-158, it is mentioned that there is an extended pipeline for data collection to build datasets for circuits with 7-10 components. What does extended pipeline mean? Are there any differences to the one used in LaMAGIC?
- There is a runtime illustration for the proposed approach. What is the runtime for baselines? It seems that the proposed method is the most time-consuming one as it introduces model inference and simulator feedback in its workflow.
- The base pretrained language model used in this work is Flan-T5-base, which is quite an old one. I acknowledge that this choice is adopted from LaMAGIC. However, the whole community of pretrained language models evolved very quickly. To make this work solid and up-to-date, new and stronger language models should be adopted (e.g., Qwen or Llama). Besides, a discussion about when base language model becomes stronger, whether the proposed challenges still exist and proposed method still work is valuable and important.
- For the experiment in Section 6.4, are the two methods compared with the same time budget? Will random generation perform better if it was given more time budget?

**Questions:**

see above.

---

> ### Author Response · Authors · 2025-12-03
>
> We thank reviewers for their valuable feedback.
>
> [W1] The extended data collection pipeline is modified from the original LAMAGIC data collection scheme. The main difference is the support for higher component data collection of the initial data.
>
> [W2] We evaluate our methods with various baselines, including sampling and filtering, and MCTS-Base. As these methods are evaluated using the same spice simulation budget, the runtime is similar to our SELECT approach.
>
> [W3] Thanks for your feedback. We will explore advanced model Qwen and Llama as future work to improve this direction.
>
> [W4] Yes, they are evaluated using the same spice simulation budget. As illustrated in Figure 5.c, the GPU computation time is marginal compared to the spice simulation (GPU time 0.2s vs. spice simulation time 10s) per circuit. Adding 2% more computation time won’t improve the performance of the random generation.

---

### Official Review · Reviewer_Tme7 · 2025-11-01

**Soundness:** 2
**Presentation:** 2
**Contribution:** 2
**Rating:** 2
**Confidence:** 5

**Summary:**

This paper proposes a framework that combines a pretrained Transformer language model with Monte Carlo Tree Search (MCTS) and circuit simulation feedback to automatically generate analog power converter topologies. Unlike prior one-shot or reinforcement-learning methods, SELECT performs search-based decoding, where the language model guides MCTS to explore only the most promising next tokens while simulator feedback evaluates and backpropagates performance rewards. Key innovations include a probability-guided UCB (P-UCB) node selection strategy and structural-token filtering to improve efficiency. Experiments show up to 435% higher success rates than prior methods and demonstrate scalability to 7–10-component circuits.

**Strengths:**

+ Novel integration of the LLM and search. SELECT is the first framework to combine a pretrained Transformer with Monte Carlo Tree Search (MCTS) for analog circuit topology generation.
+ Extending the topology generation to 7–10-component converters is an important step to make the work more practical.

**Weaknesses:**

- Overly exaggerated claim. For people who are familiar with analog circuit design, the paper made a serious overclaim of the generality of their method. Throughout the paper, experiment results are only provided for power converters, which are merely one type of analog/mixed-signal circuits. There is no evidence that directly supports or suggests that the proposed data representation or the language model formulation can be extended beyond the converter circuits. Another overclaim comes from exaggerating the accomplishment on extending the previous method from 3~5-component designs to 6~10-component designs. Fewer than 10-component analog circuits have been quite exhaustively studied by analog designers. Although a method to automate the generation of circuits with such a low component count may be of some theoretical interest, it is of little practical use.
- Lack of a clear explanation of the algorithmic method. The paper does not provide sufficient details for the reviewer to fully grasp the relationship between the language model training/pretraining and the Monte Carlo Tree Search (MCTS) method with its various selection/expansion/evaluation steps. It is also very difficult to make sense of the baseline methods being compared with in the evaluation.
- Unclear dataset construction detail. The paper never fully explains how the 7–10-component datasets were generated. For example, the number of rollouts, filtering thresholds, or simulation settings. This omission significantly limits reproducibility.
- Poor writing and organization. The writing of the paper can be improved. In many places, simple typos greatly impede readability. For instance, for the number of components in each topology, the authors sometimes use numeric forms like “345,” sometimes hyphenated ranges like “3–5,” and sometimes words like “six.” Please ensure consistent formatting throughout.

**Questions:**

Can you explain how the data representation and the training method in this paper can effectively be used for other types of analog circuits? Please give concrete examples.

Can you give examples of 6~10-component converter designs generated by the proposed method? Do the generated designs follow well-known converter topologies? Are there novel topologies in the generated design that are non-obvious to human designers?

---

> ### Author Response · Authors · 2025-12-03
>
> [Q1] Our method focuses on the decoding stage, which only requires the symbolic token sequence and the associated simulator-based feedback loop. For other analog circuit families such as op-amps, SELECT can directly decode over their AnalogGenie-style Eulerian representations and evaluate candidates using the appropriate AC/DC simulation objectives. However, because AnalogGenie does not release its evaluation metrics, reproducing their experiments appears not feasible.
>
> [Q2] There are various topologies being generated, and they satisfy the evaluation metrics as set up in the pipeline. Whether they are truly novel and advantageous will require expert examination.

---

### Official Review · Reviewer_MukN · 2025-11-01

**Soundness:** 3
**Presentation:** 3
**Contribution:** 2
**Rating:** 4
**Confidence:** 3

**Summary:**

This paper introduces SELECT, a search-enhanced language model framework that combines transformer-based decoding with simulator-guided Monte Carlo Tree Search to improve analog circuit topology generation, demonstrating substantially higher success rates and scalability compared to prior methods.

**Strengths:**

- The proposed MCTS-based approach demonstrates superior performance in topology generation experiments, representing a promising direction for future research in this field.
- The experimental evaluation is comprehensive and well-documented.

**Weaknesses:**

- Figure 2 appears to have been adapted directly from LaMAGIC; it would be more appropriate for the authors to redraw the figure to ensure originality and consistency with their own work.
- The proposed method seems less scalable than prior approaches, as it is only applied to power converter circuit topology generation. Power converters are not necessarily the most representative or critical circuits in analog research. The authors are encouraged to discuss or extend their method to other types of analog circuits.
- Compared to one-shot generation methods, MCTS is computationally less efficient and requires numerous simulation runs, which may limit its practicality for real-world applications.

**Questions:**

- In Figure 1, why are there no collected topologies for circuits with seven devices?

---

> ### Author Response · Authors · 2025-12-03
>
> We thank the reviewer for the constructive feedback and for highlighting the strengths of our evaluation and methodology. We address each point below.
>
> [W1] We appreciate the concern about Figure 2. Since the problem setup is the same as LaMAGIC, the figure was cited from LaMAGIC to depict a standard power-converter topology and its graph representation. We will replace it with a fully updated and clearly original version in the camera-ready paper.
>
> [W2] Our method targets the decoding process, and LaMAGIC’s formulation is used only as a baseline template to illustrate how SELECT is able to scale the text-based topology generation. Under this formulation, we successfully extended generation from 3–6 component converters to 7–10 components. The approach is not tied to LaMAGIC, furthermore it is compatible with AnalogGenie-style formulations (e.g., targeting op-amp circuits). We are making efforts to implement the SELECT algorithm on AnalogGenie dataset, however, because AnalogGenie does not release its evaluation framework, reproducing their experiments appears not feasible.
>
> [W3] While MCTS does require more simulations than one-shot methods, its guided exploration yields substantially higher success rates and scalability, as shown in our results. In practice, the computational cost is manageable because (i) SELECT prunes low-probability branches through P-UCB and structural-token filtering, and (ii) surrogate predictors can replace full simulations to reduce runtime further. Thus, the added search cost is offset by significantly improved quality and remains practical for real-world design settings.
>
> [Q1] Figure 1 summarizes topologies reported in prior work. Existing datasets do not include seven-device or more complex converter topologies. SELECT is, to our knowledge, the first framework that can explore and generate 7–10-device topologies under this performance-driven setting, so no such examples appear in the figure.

---

### Meta-Review · Area_Chair_jTaA · 2026-01-06

**Summary:**

This submission proposes SELECT, a search-enhanced LM framework for analog circuit topology generation that integrates a pretrained Transformer decoder with simulator-guided MCTS, including structural token pruning and P-UCB node selection to improve search efficiency. Reviewers agree the direction is promising and the empirical results on power converter topology generation show large relative gains in success rate over prior search baselines at matched simulation budgets.

However, the forum discussion surfaces several high-impact issues that remain insufficiently resolved after author responses:
- Scope/generalization and overclaiming: The experiments are restricted to power converters, and multiple reviewers (including a high-confidence domain expert) argue the paper overstates generality and practical impact (e.g., “complex” circuits defined by 7–10 components). The authors acknowledge wording overstated but do not provide new evidence of applicability beyond converters.
- Reproducibility and dataset construction for 7–10 components: A key negative review highlights missing details on how 7–10 component datasets were generated (rollouts, thresholds, simulation settings). The responses remain high-level (“extended pipeline,” “same SPICE budget,” “functional and isomorphic filtering”), leaving the strongest reproducibility concern largely outstanding.
- Positioning/novelty relative to existing MCTS+LM planning literature: One reviewer requests clearer differentiation from prior MCTS-LLM work (e.g., code generation/planning), as well as systematic positioning relative to analog-circuit-generation literature. The response mentions domain-specific token filtering but does not fully articulate what is fundamentally new beyond applying established search-with-priors in this domain.

Given one strong reject (2, confidence 5) plus multiple borderline-below-threshold reviews with high confidence (4s with confidence 4–5), and no reviewer score updates indicating concerns were alleviated, I recommend rejection.

**Reviewer Concerns:**

## Concerns largely addressed by rebuttal
- Figure originality / reuse: One reviewer flagged Figure 2 as adapted from LaMAGIC; authors state it was cited due to identical setup and commit to redrawing an original figure in the camera-ready.
- Runtime fairness / time budgets: Authors clarify baselines are evaluated under the same SPICE simulation budget, and argue GPU inference overhead is negligible compared to simulation time. This resolves the “is SELECT unfairly given more time” question in principle.
- Why no 7-device examples in Figure 1: Authors clarify Figure 1 summarizes prior datasets which did not include 7+ device converter topologies; SELECT explores beyond existing data.

## Concerns still outstanding
- Generality beyond power converters and practical usefulness of 7–10 components: Multiple reviewers question whether results on power converters support claims about “analog topology generation” broadly. Authors assert compatibility with AnalogGenie-style formulations but also admit they cannot reproduce AnalogGenie due to missing evaluation backend/metrics. No new experiments or convincing evidence are provided for other circuit families (e.g., op-amps), and a domain-expert reviewer argues that <10-component analog circuits are already well studied and may be of limited practical value. The authors partially soften the “complex” wording, but the broader generality concern remains.
- Dataset construction and reproducibility for 7–10 component regime: The most critical unresolved issue is how the 7–10-component datasets were generated, including number of rollouts, filtering thresholds, and simulation settings. The response does not provide the level of detail needed to reproduce results or assess potential selection bias. This is central because the headline claim relies on scaling to 7–10 components.
- Novelty/positioning relative to prior MCTS-LLM and analog generation work: Reviewers ask for systematic comparison/clarification: (i) what prior analog generation systems can handle in terms of complexity, (ii) what is new beyond “MCTS guided by LM priors,” and (iii) how domain-specific adaptations (P-UCB, structural token pruning) differ from established MCTS-LLM planning/code-generation methods. Authors mention p-filtering and P-UCB but do not provide a clear conceptual delineation of novelty nor a strong literature positioning beyond stating incompatibilities with non-open evaluation frameworks.
- Definition of “complex” and design-space claims: Reviewers request context for calling 8–10 components “complex,” and ask what portion of enumerated topologies are viable and how they are filtered. Authors acknowledge the term “complex” was overstated and redefine it relative to one-shot methods’ capabilities, but the paper still lacks a convincing design-space analysis.
- Qualitative validity and novelty of generated circuits: One reviewer asked for examples of generated 6–10-component converters and whether they match known topologies or represent novel, non-obvious designs. Authors respond that novelty requires expert examination, but do not provide concrete examples or analysis, leaving the practical design relevance unclear.
- Model choice and modernity of LM backbone: A reviewer notes the LM is Flan-T5-base and asks whether a stronger modern LM would change the conclusions. Authors defer to future work; the concern remains, especially because “limited pretrained LM capability” is part of the motivation for adding search.

**Reviewer Scores:**

- `MukN (4, confidence 3): likely 4 → 4 .` Some issues were addressed, which could nudge upward slightly. But the scope/generalization concern and compute practicality remain, so a strong upward move is unlikely.
- `Tme7 (2, confidence 5): likely 2 → 2.` Their core objections remain only partially answered. No new evidence for broader generality was provided.
-` ahJq (4, confidence 4): likely 4 → 4 .` Runtime fairness was clarified; however, the backbone-model modernity question and dataset/pipeline detail remain open. Discussion might improve alignment but not decisively.
- `v82m (4, confidence 5): likely 4 → 4.` This reviewer’s concerns about definition of complexity, design-space justification, MCTS-LLM positioning, and reproducibility are not fully resolved. While the authors acknowledge overstatement and claim code is open-sourced, missing dataset construction specifics and positioning remain.

Overall, even with full discussion, the panel would likely converge to a mixed view with at least one high-confidence reject and multiple borderline concerns unchanged.

---

### Decision · Program_Chairs · 2026-01-26

Reject